# Targeting *BRAF* Activation as Acquired Resistance Mechanism to EGFR Tyrosine Kinase Inhibitors in *EGFR*-Mutant Non-Small-Cell Lung Cancer

**DOI:** 10.3390/pharmaceutics13091478

**Published:** 2021-09-15

**Authors:** Frank Aboubakar Nana, Sebahat Ocak

**Affiliations:** 1Institut de Recherche Expérimentale et Clinique (IREC), Pôle de Pneumologie, ORL et Dermatologie (PNEU), Université catholique de Louvain (UCLouvain), 1200 Brussels, Belgium; frank.aboubakar@uclouvain.be; 2Division of Pneumology, Cliniques Universitaires St-Luc, UCLouvain, 1200 Brussels, Belgium; 3Division of Pneumology, CHU UCL Namur (Godinne Site), UCLouvain, 5530 Yvoir, Belgium

**Keywords:** *EGFR*-mutant NSCLC, *BRAF* mutation, *BRAF* fusion, targeted therapy, resistance mechanisms, osimertinib, RAS/MAPK, BRAF TKI, MEK TKI

## Abstract

Osimertinib has become a standard of care in the first-line treatment of advanced-stage non-small-cell lung cancer (NSCLC) harboring exon 19 and 21 activating mutations in the *EGFR* gene. Nevertheless, the 18.9-month median progression-free survival emphasizes the fact that resistance to osimertinib therapy is inevitable. Acquired resistance mechanisms to osimertinib in EGFR-driven NSCLC include *MET* amplification, *EGFR* C797S mutation, neuroendocrine differentiation, small-cell lung carcinoma histologic transformation, *PD-L1* and *KRAS* amplifications and *ESR1-AKAP12* and *MKRN1-BRAF* translocations, as well as *BRAF* V600 mutation. This last one represents 3% of the acquired resistance mechanisms to osimertinib. In this review, we discuss the rationale for EGFR/BRAF/MEK co-inhibition in the light of a clinical case of *EGFR*-mutant NSCLC developing a *BRAF* V600 mutation as an acquired resistance mechanism to osimertinib and responding to the association of osimertinib plus dabrafenib and trametinib. Additionally, we discuss the acquired resistance mechanisms to osimertinib plus dabrafenib and trametinib combination in that context.

## 1. Introduction

Lung cancer is the leading cause of cancer death and the second most commonly diagnosed cancer worldwide [1]. Histologically, lung cancer, arising from lung epithelial cells, is divided into two main types, non-small-cell lung cancer (NSCLC) and small-cell lung cancer (SCLC), which represent 85% and 15% of all lung cancers, respectively [2]. Over the last 30 years, advances in next-generation sequencing (NGS) technology have led to vast improvements in our understanding of the molecular biology that underpins NSCLC, including identification of key and potentially targetable genetic aberrations. A molecular classification of NSCLC has emerged, besides histological classification, leading to new biological insights and targeted therapies directed towards specific molecular abnormalities found in small subsets of NSCLC, such as mutations in epidermal growth factor receptor (*EGFR*) and B-Raf proto-oncogene (*BRAF*) genes or translocations in anaplastic lymphoma kinase (*ALK*), Ret proto-oncogene (*RET*), or C-Ros-1 proto-oncogene (*ROS1*) genes [3,4]. *EGFR* mutation is the leading genetic alteration in NSCLC, accounting for 12% of unselected lung adenocarcinomas (LUADs) and 44% of LUADs in never-smokers in the Caucasian population. Exon 19 deletion and exon 21 L858R sensitizing *EGFR* mutations constitute 86% of *EGFR* mutations in NSCLC and confer high responsiveness (sensitivity) to EGFR tyrosine kinase inhibitors (TKIs) [5]. TKIs targeting *EGFR* activating mutations significantly improved the outcome of *EGFR*-mutant NSCLC. Recently, a third-generation TKI, osimertinib, improved progression-free survival (PFS), overall survival (OS) and intracerebral efficacy in comparison to first-generation EGFR TKIs gefitinib or erlotinib in advanced-stage *EGFR*-mutant NSCLC [6,7] (Appendix A). Moreover, osimertinib displayed a safety profile similar to first-generation TKIs [6]. Therefore, osimertinib has become the standard-of-care (SOC) in the first-line treatment of advanced-stage *EGFR*-mutant NSCLC.

Unfortunately, despite these remarkable outcome improvements in *EGFR*-mutant NSCLC treated with an EGFR TKI, almost all patients inexorably develop progressive disease during TKI treatment [7]. Tumor progression is driven by acquired resistance mechanisms (ARMs) to EGFR TKIs, which are quite heterogenous molecular alterations. These ARMs to EGFR TKIs can be broadly grouped into EGFR-dependent (*EGFR* C797S mutation, *EGFR* amplification) or EGFR-independent mechanisms (bypass signaling, downstream pathway activation, or histologic and phenotypic transformation) [8]. The management of *EGFR*-mutant NSCLC with ARMs to osimertinib presenting multiple targetable alterations is still unclear. The molecular heterogeneity related to ARMs to EGFR TKIs emphasized the weakness of therapeutic options after progression on osimertinib. Platinum doublet chemotherapy-based regimens are recommended in patients who progress on osimertinib. Nevertheless, some EGFR-dependent ARMs, such as the *EGFR* C797S mutation, seem to be targetable with new fourth-generation EGFR TKIs in preclinical studies (e.g., BLU-945 [9,10] and EAI045 [11]) and a clinical trial is currently evaluating the efficacy and safety of one of them (BLU-945) in *EGFR*-mutant NSCLC (Appendix A). Interestingly, some EGFR-independent ARMs to osimertinib, such as Hepatocyte Growth Factor Receptor (*MET*) or *HER2* amplification and *BRAF* activation, are potentially targetable with third- or fourth-generation EGFR TKIs combined with targeted therapies against *MET* amplification, *HER2* amplification, or *BRAF* activation. In this review, we discuss *BRAF* activation as an EGFR-independent ARM to osimertinib. We also tackle how we can use EGFR/BRAF/MEK co-inhibition as a new treatment strategy in *BRAF* activation as an ARM to osimertinib.

## 2. *BRAF*-Mutant NSCLC

*BRAF* is a member of the RAS/mitogen-activated protein kinase (MAPK) signaling pathway that mediates cell growth and malignant transformation [12,13,14] (Figure 1). The incidence of *BRAF* mutation is the highest in melanoma, observed in approximately one half of the cases [15]. The incidence is lower in NSCLC, with *BRAF* mutations observed in 2–6% of advanced-stage LUADs [3,16,17] (Table 1). In contrast to *EGFR* mutations in NSCLC, which occur mainly in never-smokers, *BRAF* mutations are more frequent in former or current smokers [16] (Table 1). The role of tobacco in *BRAF* mutagenesis is also supported by the fact that the spectrum of *BRAF* mutations is limited to transversion mutations [16]. Importantly, advanced-stage *BRAF*-mutant LUADs naïve from any treatment do not harbor *EGFR* or *KRAS* co-mutations; *BRAF* V600, *EGFR* and *KRAS* mutations are mutually exclusive in this setting. However, *BRAF*-mutant LUADs can occur concomitantly to other genomic alterations, such as Phosphatidylinositol-4,5-Bisphosphate 3-Kinase Catalytic Subunit Alpha (*PIK3CA*) mutations or *ALK* translocations [16]. The most common *BRAF* mutation is the valine (V) to glutamate (E) substitution at residue 600 (V600E) located in exon 15 of the gene, which constitutively maintains BRAF in an active state [14,16,17]. Conversely to wild-type *BRAF*, the constitutively active *BRAF* V600 mutation does not require protein homodimerization to switch toward the active state [13,14]. The *BRAF* V600 mutation accounts for 50% of all *BRAF* mutations in advanced-stage lung cancer at diagnosis (Table 1) and acts as an addictive oncogenic driver in LUAD [12,16,17] (Figure 1).

## 3. *BRAF* Alterations as ARMs to Osimertinib in *EGFR*-Mutant NSCLC

In vitro and in vivo data showed that the RAS–MAPK pathway activation, through an *NRAS* mutation, for instance, leads to acquired resistance to EGFR TKIs [24]. Activation of BRAF can occur as an ARM to osimertinib in advanced-stage NSCLC harboring the *EGFR* T790M mutation [8] (Figure 1). *BRAF* mutations [8,25] and *BRAF* rearrangements [8,26,27] (Figure 2) are the two genomic alterations leading to *BRAF* activation as an EGFR-independent ARM to osimertinib. EGFR-dependent ARMs encompass *EGFR* amplifications and mutations, such as the *EGFR C797X* mutation. EGFR-independent ARMs involve bypass signaling activation (*MET*, *HER2* and *FGFR* amplifications, as well as *ALK*, *ROS1* and *RET* rearrangements), epithelial-to-mesenchymal transition (EMT) through AXL receptor tyrosine kinase (AXL) activation, SCLC transformation and downstream pathway activation, such as *PI3K* mutations, and RAS–MAPK pathway activation, including *BRAF* mutations and rearrangements [8,28]. Bypass signaling activation by *BRAF* activation leads, in tumor cells, to increased cell survival and proliferation, while apoptosis is decreased [29,30] (Figure 1).

Unlike *BRAF* mutations as a primary oncogenic driver in NSCLC, *BRAF* mutations as an ARM to osimertinib can occur concomitantly with EGFR mutations, which are mainly observed in never-smokers. *BRAF* V600 mutations as an ARM, specifically, occur in approximately 3% of *EGFR*-mutant NSCLCs exposed to osimertinib in the first- or second-line setting (Table 2). Non-V600 *BRAF* mutations, such as G469A, have also been reported as an ARM to osimertinib (Table 2) [29].

*BRAF* rearrangements (Figure 2) have been described in a number of different cancers, mainly (85% of the cases) in astrocytic pilocytomas (Table 1), the most common childhood brain tumor [23]. *BRAF* rearrangements as an ARM to osimertinib occur in approximatively 2% of *EGFR*-mutant NSCLCs exposed to osimertinib (Table 2), both in the first- and second-line settings [8,26,27]. Several different 5′ fusion partners for oncogenic *BRAF* exist in different cancers and vary by tumor type. Acylglycerol kinase gene (AGK) and praja ring finger ubiquitin ligase 2 gene (*PJA2*)/*BRAF* fusion partners have been described in *BRAF* rearrangements as an ARM to osimertinib. Activating *BRAF* rearrangements results in loss of the N-terminal auto-inhibitory domain of BRAF, leading to the formation of constitutively active BRAF fusion protein dimers that activate the RAS–MAPK pathway [26,27].

## 4. Targeting EGFR/BRAF/MEK Pathway in NSCLC with *BRAF* Activation as an ARM to Osimertinib

Patients experiencing tumor progression with *BRAF* genetic alteration as an ARM to osimertinib in *EGFR*-mutant NSCLC usually receive platinum-based chemotherapy. As of today, there is no therapeutic strategy using targeted therapy to inhibit both the EGFR and BRAF/MEK pathways. The management of tumors presenting multiple targetable mutations is still unclear. However, EGFR-independent ARMs include bypass signaling activation, such as *MET* and *HER2* amplifications, which are targetable. For instance, in *EGFR*-mutant NSCLC harboring *MET* amplification as ARM to EGFR TKI, dual inhibition of EGFR and MET with the combination of osimertinib and savolitinib, respectively, provided promising results with an acceptable risk-benefit profile and encouraging antitumor activity [31]. This EGFR–MET pathway co-inhibition is evaluated in an ongoing phase 2 clinical trial, SAVANNAH (NCT03778229) [32]. Understanding of ARMs to osimertinib and evaluation of targeted treatment options post-tumor progression are ongoing in a large phase 2 open-label trial, ORCHARD (NCT03944772) [33]. The combination of the EGFR–MET bispecific antibody amivantamab with the third-generation EGFR TKI lazertinib also showed a synergistic inhibition of tumor growth and a good safety profile in a cohort of *EGFR*-mutant NSCLC patients with resistance to osimertinib in a phase 1 trial, CHRYSALIS [34]. Unfortunately, targeted treatment strategies to overcome BRAF activation as an ARM to osimertinib have not been considered in clinical trials so far. In this part, we tackle targeted treatment strategies to overcome ARMs to osimertinib, with a focus on BRAF activation.

Osimertinib is SOC in advanced-stage *EGFR*-mutant NSCLC in the first- or second-line setting (in second-line, the occurrence of an *EGFR* T790M mutation of resistance to first- or second-generation EGFR TKIs is required) [6,7,35]. Several other third-generation EGFR TKIs (e.g., lazertinib, almonertinib and alflutinib) have been evaluated in clinical trials, mainly early-phase (Appendix A) and/or are in clinical development, (Appendix A). Interestingly, a fourth generation EGFR TKI, BLU-945, overcomes the *EGFR* T790M as well as the C797S mutations of resistance to first/second- and third-generation EGFR TKIs, respectively, and demonstrated interesting antitumor activity in preclinical studies [9,10]. The clinical evaluation of BLU-945 is ongoing in a phase 1/2 clinical trial (NCT04862780) (Appendix A). Furthermore, the combination of the BRAF kinase inhibitor dabrafenib with the MEK kinase inhibitor trametinib is approved by the U.S Food and Drug Administration, as well as the European Medicines Agency, in the first-line setting of advanced-stage NSCLC harboring a *BRAF* V600E mutation based on a phase 2 trial which showed an objective response rate (ORR) of 64%, a median duration of response of 15.2 months and a median PFS of 14.6 months, if assessed by the independent review committee, or 10.9 months, if investigator-assessed [36] (Appendix A). Importantly, the antitumor activity of the dabrafenib plus trametinib combination is higher than dabrafenib monotherapy, with an ORR of 67% and 33%, respectively [37]. In addition, the safety profile of the combination of BRAF plus MEK inhibitors encorafenib plus binimetinib is better compared with the BRAF inhibitor encorafenib or vemurafenib in monotherapy [38].

### 4.1. BRAF V600 Mutation

In a patient suffering from an *EGFR*-mutant NSCLC with *BRAF* V600E mutation as an ARM to osimertinib, we observed that the EGFR/BRAF/MEK pathway co-inhibition with dabrafenib, trametinib plus osimertinib triple therapy (as suggested in Figure 1) was more effective than the dabrafenib plus trametinib dual therapy [25]. Indeed, following tumor progression on osimertinib related to the occurrence of a *BRAF* V600 mutation, osimertinib was first stopped and the dabrafenib plus trametinib combination initiated. However, after six weeks of dabrafenib plus trametinib treatment, symptomatic tumor progression was observed. Therefore, it was decided to restart osimertinib while continuing dabrafenib plus trametinib. This triple inhibition led to clinical improvement and partial tumor response. This observation is consistent with preclinical data reporting that BRAF inhibitor encorafenib suppressed MEK signaling but had no significant effect on ERK phosphorylation, while the combination of encorafenib and osimertinib significantly reduced both MEK and ERK phosphorylation, as well as growth of NSCLC cells harboring both *EGFR* and *BRAF* V600E mutations [30].

In our review of nine case reports using dabrafenib, trametinib plus osimertinib triple therapy for EGFR/BRAF/MEK pathway co-inhibition (as suggested in Figure 1), there was a meaningful improvement of PFS, as well as a clinical benefit [25]. Toxicities reported with the triple therapy included fatigue, pyrexia, dysgueusia, nausea, vomiting, diarrhea, pneumonitis, or creatinine phosphokinase (CPK) elevation; these adverse events were managed through dose modifications of osimertinib and/or dabrafenib and trametinib. In theory, other BRAF and MEK inhibitor combinations evaluated in melanoma [38,39] and in ongoing phase 2 clinical trials in *BRAF*-mutated NSCLC, such as encorafenib plus binimetinib in advanced-stage (NCT03915951) (Appendix A), or vemurafenib plus cobimetinib in the neoadjuvant and adjuvant setting (NCT04302025), may also be combined with other third-generation EGFR TKIs (e.g., lazertinib, almonertinib, alflutinib) or the fourth generation EGFR TKI BLU-945. This last one has the advantage to target both acquired *EGFR* T790M and C797S mutations. Nevertheless, safety evaluation in clinical setting is required and ongoing.

### 4.2. Non-BRAF V600 Mutations and BRAF Rearrangements

Non-V600 *BRAF* mutations, such as *BRAF* G469A, are other *BRAF* activation mechanisms which confer acquired resistance to osimertinib through the RAS–MAPK pathway activation. Importantly, in vitro data showed that the combination of osimertinib with a MEK inhibitor (selumetinib or trametinib) induced a synergistic effect, resulting in the suppression of proliferative pathways and the induction of apoptotic signaling, thus overcoming osimertinib resistance [29]. These preclinical data pointing out the efficacy of the dual inhibition of the EGFR–MEK pathway need to be tempered by the fact that all clinical data obtained so far favored the combination of BRAF and MEK inhibitors compared with BRAF inhibitors alone in terms of antitumoral efficacy, as well as toxicity profile [37,38,39].

*BRAF* activation through *BRAF* rearrangement is another ARM to osimertinib in *EGFR*-mutant NSCLC [26]. Importantly, combined inhibition of BRAF and EGFR with vemurafenib or dabrafenib plus osimertinib inhibited growth of isogenic cell line models and of a patient-derived cell line harboring a *BRAF* rearrangement and an *EGFR* mutation in a synergistic manner (Figure 1).

## 5. Discussion and Perspectives

The major clinical obstacle that limits the long-term benefit of osimertinib in patients with *EGFR*-mutant NSCLC is the emergence of ARMs [6,8,28]. The SOC for patient progressing on osimertinib remains platinum-based chemotherapy, but the results are disappointing. More effective treatments targeting molecular alterations to overcome ARMs to osimertinib are needed to improve the outcome of these patients. The targeted therapy approach may have the advantages of a better safety profile and a better outcome and quality of life than chemotherapy, as reported in first- and second-line settings in *EGFR*-mutant NSCLC [35,40,41]. The characterization of acquired molecular alterations to osimertinib is quite well elucidated. A targeted therapy trial, based on multiple molecular alterations, is ongoing with the combination of the third generation EGFR TKI lazertinib and the EGFR-MET bispecific antibody amivantamab (NCT04077463). This combination is expected to be effective in ARMs to osimertinib involving *EGFR* or *MET* amplifications. The combination of osimertinib and savolitinib, evaluated in the ongoing phase 2 SAVANNAH trial (NCT03778229) [32] and the phase 2 open-label ORCHARD trial (NCT03944772) [33], is also expected to be effective in *EGFR*-mutant NSCLC with *MET* amplification, while the combination of osimertinib and necitumumab is expected to be effective in *EGFR*-mutant NSCLC with *EGFR* amplification. Osimertinib plus selpercatinib and osimertinib plus alectinib could potentially be effective in *EFGR*-mutant NSCLC with *RET* and *ALK* rearrangements, respectively. Unfortunately, the targeted therapy approach against mechanisms leading to BRAF activation, which include V600 and non-V600 *BRAF* mutations and *BRAF* translocations which account for approximatively 5% of ARMs to osimertinib, has not been explored in clinical trials yet.

This BRAF activation in *EGFR*-mutant NSCLC seems to be exclusively reported in patients treated with osimertinib [8], which has become the SOC first-line treatment in patients suffering from an advanced-stage NSCLC harboring common exon 19 and 21 *EGFR* mutations. The proportion of BRAF activation may potentially increase in the context of tumor progressing on osimertinib in the adjuvant setting [42]. Furthermore, BRAF activation as ARM is expected to appear with other third generation EGFR TKIs in the pipeline, such as lazertinib, almonertinib and alflutinib. Preclinical and clinical data from case reports suggest that the targeted therapy approach using the triple EGFR/BRAF/MEK pathway co-inhibition may be effective and well tolerated in patients with *EGFR*-mutant NSCLC and BRAF activation as an ARM to EGFR TKIs [25,26,27,29,30,42]. Therefore, clinical trials evaluating the EGFR/BRAF/MEK pathway co-inhibition in patients with BRAF activation as ARM to EGFR TKIs are needed to assess the optimal dose of the combined drugs which is effective and has a good toxicity profile in the whole population. At the end, it would be interesting to assess whether the EGFR/BRAF/MEK pathway co-inhibition ultimately alters the natural history of these tumors analogously to ALK inhibitors in *ALK*-rearranged NSCLC. To do so, plasma genotyping needs to be considered during the treatment, as well as tumor biopsy at progression to detect the ARM to the EGFR/BRAF/MEK pathway co-inhibition. Interestingly, an ARM to EGFR/BRAF/MEK pathway co-inhibition, with osimertinib as the EGFR TKI, may be the *EGFR* C797S mutation, which is potentially targetable with the fourth generation EGFR TKI BLU-945 [9,10,25,43].

## Figures and Tables

**Figure 1 pharmaceutics-13-01478-f001:**
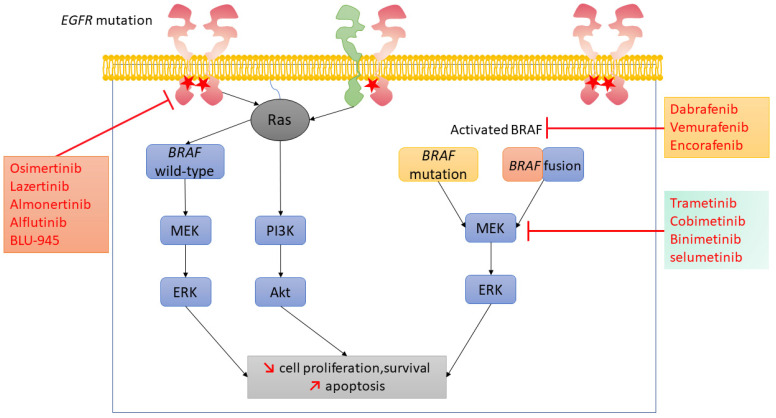
Triple targeted therapy suggestions in tumor cells harboring *EGFR* mutation and *BRAF* activation. Mutation in the *EGFR* gene leads to the EGFR receptor homo- or hetero-dimerization, which results in the activation of downstream effectors, including Ras–MAPK (with wild-type *BRAF* in blue) and PI3K-Akt, leading to increased cell proliferation and survival and decreased apoptosis. In a tumor cell harboring an *EGFR* mutation and a *BRAF* genetic alteration (mutation in yellow or fusion in blue/orange), BRAF can directly activate MEK, bypassing the Ras signaling and, therefore, leading to resistance to osimertinib. The combination of EGFR tyrosine kinase inhibitors (TKIs) and *BRAF* plus MEK kinase inhibitors induces EGFR/BRAF/MEK co-inhibition, which overcomes BRAF-acquired resistance mechanism (*BRAF* mutation or fusion) to third-generation EGFR TKIs. Abbreviations: Akt, proto-oncogene c-Akt; BRAF, B-Raf proto-oncogene; EGFR, epidermal growth factor receptor; ERK, mitogen-activated protein kinase 2; MEK, mitogen-activated protein kinase 1; PI3K, phosphatidylinositol 3-kinase; Ras, ras sarcoma viral pro-to-oncogene; MAPK, mitogen activated protein kinase.

**Figure 2 pharmaceutics-13-01478-f002:**
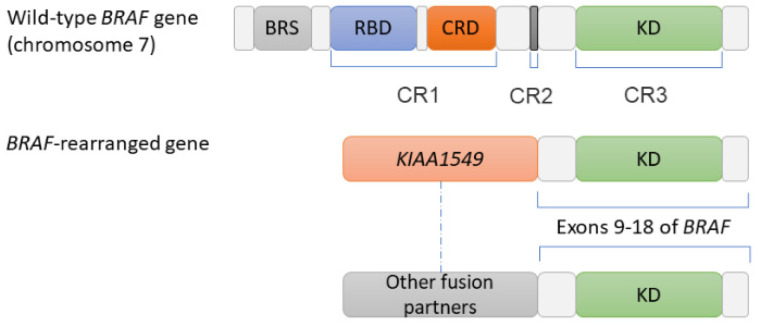
Schematic representation of the wild-type and the rearranged *BRAF* gene. All *BRAF* rearrangements lead to a break leaving the serine/threonine kinase domain (or conserved region 3) and the ATP binding pocket of *BRAF* intact, while the N-terminal domains (conserved region 1 (CR1) and conserved region 2 (CR2)) are lost, replaced by the fusion partner. Since CR1 is known to auto-inhibit the kinase domain of the BRAF protein, its loss constitutively activates the kinase and explains the oncogenic potential of the rearrangement. *KIAA1549* is the most frequent *BRAF* fusion partner (associated with pilocytic astrocytoma), but 40 other partners have also been described (https://ccsm.uth.edu/FusionGDB/, (accessed on 20 August 2021)), such as *AGK* and *MKRN1* (reported as acquired resistance mechanisms to EGFR tyrosine kinase inhibitors in *EGFR*-mutant lung adenocarcinoma). Abbreviations: AGK, AcylGlycerol kinase; BRS, BRAF-specific region; CR1-3, conserved region 1-3; CRD, Cys-rich domain; KD, kinase domain; MKRN1, Makorin ring finger protein 1; RBD, Raf-like Ras-binding domain.

**Table 1 pharmaceutics-13-01478-t001:** *BRAF* gene alterations in untreated non-small-cell lung cancer (NSCLC) and pilocytic astrocytoma.

Cancer Types	BRAF Alterations	Proportions	References
NSCLC	*BRAF* mutations	2–6% of NSCLC	[3,18,19]
	V600E	50–65% of *BRAF* mutations	[19,20,21]
	Non-V600E	35–50% of *BRAF* mutations	[19,20,21,22]
	In never smokers	19–23% of *BRAF* mutations	[16,19]
	In former smokers	69–72% of *BRAF* mutations	[16,19]
	In current smokers	5–13% of *BRAF* mutations	[16,19]
	BRAF rearrangements	0 (not reported)	[19]
Pilocytic astrocytoma	*BRAF* rearrangements	78.13%	[19,23]

**Table 2 pharmaceutics-13-01478-t002:** *BRAF* gene alterations as acquired resistance mechanisms to osimertinib in *EGFR*-mutant non-small-cell lung cancer.

*BRAF* Gene Alterations		Proportions	References
Mutations	V600E	3%	[8]
	Non-V600E	0 (not reported)	[29]
Rearrangements	AGK, MKRN1, PJA2 as fusion partners	1–2%	[8,26,27,28]

## Data Availability

The data presented in this review are openly available in the reference section as well as Appendix A.

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
