# Peer review of "Targeting BRAF Activation as Acquired Resistance Mechanism to EGFR Tyrosine Kinase Inhibitors in EGFR-Mutant Non-Small-Cell Lung Cancer"

_pharmaceutics, 2021, doi:10.3390/pharmaceutics13091478_

Round 1

Reviewer 1 Report

In this review article, the authors focus on the epidermal growth factor receptor (EGFR) mutation-driven non-small-cell lung cancer (NSCLC) that develops a B-Raf proto-oncogene product (BRAF) V600 mutations as an acquired resistance to osimertinib, an EGFR tyrosine kinase inhibitor (TKI). This represents 3% of the acquired resistance to osimertinib. They discuss the rationale of three-drug combination therapy for EGFR-mutation-driven NSCLC patients who developed BRAF mutation after first-line treatment with osimertinib. The three drugs consist of osimertinib, dabrafenib and tramethinib, which target EGFR, BRAF and the MAPK/ERK kinase (MEK), respectively.

              While BRAF V600 mutations occurs only in 3% of the acquired resistance to the EGFR TKI osimertinib, the authors should explain why they focus on only BRAF mutations as an EGFR-independent acquired resistance to osimertinib. They should also discuss BRAF mutations as an acquired resistance to other EGFR TKI than osimertinib. As mentioned above, the authors propose the three-drug combination therapy but do not explain why three drugs against EGFR, BRAF and MEK are better than two against EGFR and BRAF. In theory, an inhibitor to BRAF should inactivate the pathway. Why is MEK redundantly inactivated by a specific inhibitor.

Specific comments

1) Figures 1 and 2 should be combined into one.

2) Consistent usages of amino acid nomenclatures throughout the manuscript, either Leu858Arg or L858R. Furthermore, EGFR L858R is correct, while EGFR L858R is not correct since it is a protein, not gene. Check this point throughout the manuscript.

3) Provide a summary Table or Figure of BRAF mutations (V600 and others with percentages and references) found in BRAF-mutant LUAD naïve from any treatment, in former or current smokers, in melanoma or NSCLC, in EGFR-independent BRAF genomic rearrangements in NSCLC and other cancers such as astrocytic pilocytomas and childhood brain tumor, and in BRAF as EGFR-independent ARMs, etc. since the authors specifically focus on BRAF mutations. Also provide a figure showing BRAF rearrangements as ARMs to osimertinib as well as in other cancers, which include fusion partners.

4) Make a Table showing first-, second-, third- and fourth-generation EGFR TKIs, BRAF KIs and MEK KIs, and explain differences amongst the generations.

Minors

5) Lines 58 and 63: (ARM) -> (ARMs).

6) Line 59: EGFR C797S -> EGFR C797S since this is a protein, not gene.

7) Line 80: Rephrase this sentence.

8) Line 86: Explain what PIK3CA is?

9) Line 97: Redundant “a member ------“.

10) Line 98: Is EGFR T790M a gene or a protein?

11) Line 99: Are BRAF gene or BRAF protein rearrangements?

12) Lines 20 and 101: BRAF V600 or BRAF V600X, or EGFR C797 or EGFR C797X.

13) Line 104: What is AXL?

14) Lines 131 and 141: Redundant abbreviations. Many of them are explained in the main text.

15) Line 160: What is EFGR?

16) Line 177: Remove the abbreviation FDA, EMA since they never appear afterwards.

17) Line 176: BRAF and MEK are Ser/Thr kinases, not tyrosine kinases. Do not use TKIs for their inhibitors.

18) A number of typos throughout the manuscript: e.g., line 180: a  median -> a median, line 184: alone( [31]. -> alone [31]; line 204: theses -> these; etc, etc.

Author Response

  1. Figures 1 and 2 should be combined into one.
  • As requested, we combined the two figures into a single figure, which we entitled “Figure 1: Triple targeted therapy suggestions in tumor cells harboring EGFR mutation and BRAF activation”.
  • The legend of this new Figure 1 has been adapted accordingly.

  1. Consistent usages of amino acid nomenclatures throughout the manuscript, either Leu858Arg or L858R. Furthermore, EGFR L858R is correct, while EGFR L858R is not correct since it is a protein, not gene. Check this point throughout the manuscript.
  • We modified “Leu858Arg” (used only once in the text) by “L858R” (used everywhere else in the text) to have a consistent designation.
  • We also verified all the gene and protein symbols and corrected the mistakes (consisting mainly in gene symbols which were not italicized).
  • In the precise case of EGFR L858R, we refer to the gene (L858 point mutation in exon 21 of the EGFR gene) and so EGFR should remain italicized.

  1. Provide a summary Table or Figure of BRAF mutations (V600 and others with percentages and references) found in BRAF-mutant LUAD naïve from any treatment, in former or current smokers, in melanoma or NSCLC, in EGFR-independent BRAF genomic rearrangements in NSCLC and other cancers such as astrocytic pilocytomas and childhood brain tumor, and in BRAF as EGFR-independent ARMs, etc. since the authors specifically focus on BRAF mutations. Also provide a figure showing BRAF rearrangements as ARMs to osimertinib as well as in other cancers, which include fusion partners.
  • To answer this concern, we created a new figure, which we entitled “Figure 2: Schematic representation of the wild-type and the rearranged BRAF gene”, as well as two tables, entitled “Table 1: BRAF gene alterations in untreated non-small cell lung cancer (NSCLC) and pilocytic astrocytoma” and “Table 2: BRAF gene alterations as acquired resistance mechanisms to osimertinib in EGFR-mutant non-small cell lung cancer”.
  • We added mentions to the new figure and tables in the text.

  1. Make a Table showing first-, second-, third- and fourth-generation EGFR TKIs, BRAF KIs and MEK KIs, and explain differences amongst the generations.
  • As requested, we made a table entitled “Supplementary Table 1: Main clinical trials performed with third-generation EGFR tyrosine kinase inhibitors and BRAF +/- MEK inhibitors in advanced-stage NSCLC”.
  • We did not include first- and second-generation EGFR TKIs in this table because the manuscript does not focus on them and they are currently almost not used in standard clinical practice (as mentioned in the manuscript, osimertinib recently became the new standard first-line treatment in advanced-stage EGFR-mutant NSCLC). Furthermore, BRAF activation as acquired resistance mechanism to EGFR TKIs in EGFR-mutant NSCLC mainly occurs in patients exposed to osimertinib.

Minors

  1. Lines 58 and 63: (ARM) -> (ARMs).

This has been corrected.

  1. Line 59: EGFR C797S -> EGFR C797S since this is a protein, not gene.

This has not been modified because we refer to the gene, not to the protein.

  1. Line 80: Rephrase this sentence.

The sentence has been rephrased to make it more clear: “The incidence of BRAF mutation is the highest in melanoma, observed in approximately one half of the cases [15]. The incidence is lower in NSCLC, with BRAF mutations observed in 2 to 6% of advanced-stage LUADs [3,16,17] (Table 1). In contrast to EGFR mutations in NSCLC which occur mainly in never-smokers, BRAF mutations are more frequent in former or current smokers [16] (Table 1)”.

  1. Line 86: Explain what PIK3CA is?

PIK3CA is phosphatidylinositol-4,5-Bisphosphate 3-Kinase Catalytic Subunit Alpha. This has been added in the text.

  1. Line 97: Redundant “a member ------“.

We deleted “a member of the RAS–MAPK pathway,” from the text.

  1. Line 98: Is EGFR T790M a gene or a protein?

It is a gene and so, it is correct to keep it italicized.

  1. Line 99: Are BRAF gene or BRAF protein rearrangements?

BRAF rearrangement is a genetic abnormality and so, it is correct to keep BRAF italicized in that context.

  1. Lines 20 and 101: BRAF V600 or BRAF V600X, or EGFR C797 or EGFR C797X.

In line, we prefer to keep BRAF V600 while in line 101, we modified EGFR C797X into EGFR C797.

  1. Line 104: What is AXL?

AXL refers to the AXL receptor tyrosine kinase. This has been added in the text.

  1. Lines 131 and 141: Redundant abbreviations. Many of them are explained in the main text.

As the previous Figure 1 and Figure 2 have been combined, we have now a single legend, which resolves the redundancy issue for the abbreviations.

  1. Line 160: What is EFGR?

It should be EGFR and not EFGR. We corrected the typo mistake.

  1. Line 177: Remove the abbreviation FDA, EMA since they never appear afterwards.

The two abbreviations have been removed.

  1. Line 176: BRAF and MEK are Ser/Thr kinases, not tyrosine kinases. Do not use TKIs for their inhibitors.

We corrected the mistake related to these inhibitors throughout the manuscript.

  1. A number of typos throughout the manuscript: e.g., line 180: a median -> a median, line 184: alone( [31]. -> alone [31]; line 204: theses -> these; etc, etc.

We corrected the typo mistakes throughout the manuscript.

Reviewer 2 Report

the paper is very interesting and explains well the rationale for EGFR / BRAF / MEK co-inhibition in the light of a clinical case of EGFR-mutant NSCLC developing a BRAF V600 mutation as an acquired resistance mechanism to osimertinib and responding to the association of osimertinib plus dabrafenib and trametinib. 

however, there are two main aspects to be evaluated

1) a summary of the phase 1,2 and possibly 3 trials studied, with clinical and preclinical data, should be made through a table.
2) the drawings are not at the height of the magazine on which you want to publish

once these two important aspects have been clarified, I will be happy to carry out a subsequent evaluation

Author Response

  1. A summary of the phase 1,2 and possibly 3 trials studied, with clinical and preclinical data, should be made through a table.
  • As requested, we made two tables including available data from phase 1, 2, and 3 clinical trials and ongoing clinical trials with EGFR, BRAF, and MEK inhibitors, which we entitled “Supplementary Table 1: Main clinical trials performed with third-generation EGFR tyrosine kinase inhibitors and BRAF +/- MEK inhibitors in advanced-stage NSCLC” and “Supplementary Table 2: Main clinical trials ongoing or planned with third- or fourth-generation EGFR tyrosine kinase inhibitors and BRAF + MEK inhibitors in advanced-stage NSCLC”.
  • We preferred to put them in the Supplementary Material because they are large and also because they include data on each of the discussed inhibitors but not on the combination of them since there is currently no available clinical data with this combination of third-generation EGFR and BRAF + MEK inhibitors.

  1. The drawings are not at the height of the magazine on which you want to publish.

We modified the drawings to improve their quality.

Round 2

Reviewer 2 Report

the corrections requested have been made correctly.Paper turns out to be more linear and smooth. At line 90 it would be useful to insert another reference about the PIK3CA BRAF KRAS and EGFR mutations. Suggest the following work: "Molecular and Clinical Insights in Malignant Brenner Tumor of the Testis With Liver Metastases: A Case Report" Front. Oncol., 12 April 2021, Parcesepe et al.